# LitReview Arena: Evaluating Literature Review Agents with Battle-Style Peer Review Platform

**Ruotong Zhao** [1]  **Zhiyu Chen** [2]  **Xurui Liu** [1]  **Haidong Xue** [3]  **Dong Liang** [4]  **Jigao Fu** [3]  **Yanbiao Wu** [5]
**Yuanyi Zhen** [2]  **Fengli Xu** [1]  **Yong Li** [1]

## Abstract

Literature reviews are essential to scientific progress, but rigorously evaluating automatically generated reviews remains difficult because many aspects of research utility depend on expert judgment rather than reference-overlap metrics. We introduce LitReview Arena, a battle-style evaluation platform with a structured protocol tailored to literature review quality: domain experts with AI paper-writing experience compare anonymized drafts, are matched to topics within their expertise, and provide dimension-wise outcomes over five literature-review-specific criteria. From this protocol, we collect approximately 3k expert judgments, each containing five dimension-wise outcomes, and show that even the strongest current systems win only 23.0% of decisive matches against human drafts on overall utility, while agentic LLMs such as Sonar Deep Research substantially outperform base language models by over 60%. We further find that existing LLM-as-a-judge methods are substantially misaligned with human experts (Spearman's $\rho \approx 0.467$), especially on synthesis-heavy criteria such as paper structure and research suggestions. Using the collected preference data, we provide an expert-calibrated evaluator, LitJudge, which improves alignment to $\rho \approx 0.78$, comparable to inter-expert consistency; code and data are publicly available at https://github.com/VanellopeAsher/LitReview-Arena.

[1]Tsinghua University, Beijing, China [2]Zhongguancun Academy, Beijing, China [3]Zhongguancun Institute of AI, Beijing, China [4]Huazhong University of Science and Technology, Wuhan, China [5]Shanghai Institute of Microsystem and Information Technology, Shanghai, China. Correspondence to: Ruotong Zhao <zhaort22@mails.tsinghua.edu.cn>.

*Proceedings of the $43^{rd}$ International Conference on Machine Learning*, Seoul, South Korea. PMLR 306, 2026. Copyright 2026 by the author(s).

## 1. Introduction

Scientific literature reviews condense a fast-moving field into a usable mental model. They shape how researchers categorize methods, what evidence they treat as settled, and which gaps are worth pursuing next. As deep research systems improve, automated survey generation is no longer a speculative goal; the bottleneck is increasingly evaluation. Existing work has progressed in stages. Early evaluation systems emphasize literature retrieval and organization, but do not provide an evaluation mechanism for the review itself (Tang et al., 2024). More recent benchmarks and systems expand from retrieval to end-to-end review generation (e.g., DeepResearch Bench; Du et al., 2025), yet their evaluations are typically based on static reports, fixed rubrics, or proxy signals such as reference correctness, coverage heuristics, and overlap against a reference document (Tang et al., 2024; Du et al., 2025). These signals matter, but they do not resolve the central challenge in literature review assessment: many of the criteria that determine research utility are non-verifiable and inherently intuition-based. In a good review, structure is not a matter of formatting; it is an argument about what the field is and how its subareas relate. Likewise, strong gap and direction sections rarely come from generic templates. They require interpreting what is missing, what assumptions are fragile, and what experiments would resolve uncertainty. Without subjective, dynamic human preference as a first-class signal, evaluation struggles exactly where the review matters most to researchers.

Arena-style evaluation emerges as an effective approach to address this gap. LMArena demonstrated that arena votes can yield stable rankings for open-ended generation tasks that are difficult to score with absolute metrics (Zheng et al., 2023; Liu et al., 2023; Chiang et al., 2024). In literature-grounded scientific settings, SciArena extends this paradigm to non-verifiable tasks by using a battle platform and preference voting (Zhao et al., 2025). However, its target is generalized scientific, literature-grounded tasks and cannot provide a specialized, structured, end-to-end protocol tailored to literature review quality. We argue that literature review evaluation requires specialized improvements in pro-

tocol design to make expert preference data both reliable and reproducible: (1) Annotator: we recruit domain experts and constrain participation to researchers with AI paper-writing experience, rather than open enrollment, because assessing review structure and research directions depends on field-level judgment. (2) Query selection: we derive topics from real, highly cited surveys to ensure the underlying research questions are valuable and recognizable to the community, rather than synthetic prompts with unclear academic relevance. (3) Expertise matching: we precisely match expert annotators and queries, assigning each topic to annotators within their areas of expertise, so that voters evaluate reviews in domains they can genuinely peer review. (4) Structured dimensions: we define five dimensions tailored to literature review assessment: D1-literature coverage, D2-claim support, D3-paper structure, D4-research suggestions quality, and D5-overall utility, so that both citation-facing criteria and the non-verifiable dimensions that dominate research utility are explicitly captured (Yan et al., 2025). Putting these together, our protocol is designed to simulate a peer review system that is widely recognized and used in academia, while retaining the operational simplicity of arena voting. This is the key axis on which we specialize beyond SciArena: the battle platform is not only a scaling device, but a mechanism to enforce scientific rigor for a single task family (literature reviews) with a complete, structured evaluation protocol.

Based on this protocol, we introduce the LitReview Arena platform and release LitReviewBench, an arena-grounded expert preference benchmark for literature review agents. We collect a large-scale dataset of 3k expert judgments, each containing five dimension-wise outcomes, and conduct, to our knowledge, the first systematic study of how far current models are from human-level literature review performance under expert preference evaluation. We find that even the most advanced models remain far from human-level performance on overall utility (winning rate = 23.0% in decisive matches against human drafts). Paper Structure (D3) and Research Suggestions (D4) trail human experts by roughly 200 points, indicating that coherent landscape organization and non-obvious gap identification remain difficult even for strong systems. We also observe a consistent system-level trade-off: agentic models substantially outperform pure language models across dimensions, but at a substantial test-time computing cost—on average consuming roughly $15\times$ the token budget for the same topic. To make expert-aligned evaluation practical at low ongoing cost, we release a low-cost, expert-aligned evaluator for offline evaluation. Uncalibrated LLM-as-a-judge methods are substantially misaligned with expert preferences, producing leaderboards that disagree with expert rankings. Using LitReviewBench as a calibration signal, our evaluator improves expert-ranking alignment from $\rho \approx 0.467$

to $\rho \approx 0.792$ on D5, enabling scalable offline assessment and providing a training signal that can be used to improve literature review agents with expert preference supervision.

**Our contributions are as follows.**

- We propose the LitReview Arena platform together with an evaluation protocol that emphasizes fairness and scientific rigor: battle-style blind evaluation, expert-only voting, topic–expertise matching, and structured evaluation that support diagnostic analysis.
- We collect a large-scale dataset of 3k expert judgments, each containing five dimension-wise outcomes, and systematically evaluate model literature review quality against human drafts under expert preference. We find that non-human systems win only 23.0% of decisive matches against human drafts on overall utility, with the largest gaps on paper structure and research suggestions (roughly 200 points), while agentic models outperform pure language models at an average $15\times$ token cost.
- We release a low-cost, expert-aligned evaluator for offline evaluation. Using LitReviewBench as a calibration signal, it improves expert-ranking alignment from $\rho \approx 0.467$ to $\rho \approx 0.792$ on D5, enabling scalable offline assessment and providing a training signal for improving literature review agents with expert preference supervision.

**Conflict of Interest Disclosure.** The authors declare no financial conflicts of interest.

## 2. Related Work

### 2.1. Automated Literature Review Generation Systems

A growing line of work treats literature review writing as an end-to-end retrieval-and-synthesis problem: given a topic, the system collects a paper set, organizes it into an outline, and produces a narrative with citations. For example, AutoSurvey focuses on generating survey-style drafts from retrieved literature, often centering evaluation on reference generation and overall coherence (Wang et al., 2024). SurveyForge studies practical heuristics for survey writing—notably outline construction and memory-driven generation—and proposes multi-dimensional evaluations that still largely rely on operationalizable criteria (Yan et al., 2025). SurveyX frames academic survey generation as a pipeline that combines retrieval, clustering, and section-level generation (Liang et al., 2025). Other agents explore more explicit knowledge organization, such as building multiple lightweight knowledge graphs (minigraphs) and aggregating them into a review (Zhang et al., 2025).

Despite real progress in coverage and fluency, these sys-

tems are typically positioned closer to paper search and integration tools than to expert-level synthesizers. In practice, they often under-specify what makes a review usable to researchers: a defensible global structure (what belongs together and why) and a gap narrative that goes beyond generic future work templates. As a result, improvements in retrieval quality and citation formatting do not necessarily translate into better scholarly synthesis, especially when the target is to help readers reason about the field's organizing axes and the non-obvious directions that follow.

## 2.2. Static Benchmarks for Literature Review Quality

Benchmarking literature review agents has lagged behind system building, partly because expert-valued qualities are hard to reduce to deterministic checks. Existing benchmarks and datasets therefore skew toward what can be measured reliably at scale: topical relevance, summary similarity, citation matching, and claim-level factuality (Kasanishi et al., 2023; Ajith et al., 2024). Several efforts explicitly target literature-grounded generation and evaluation (e.g., SciReviewGen, SurGE, ReportBench, and related tracks), but the dominant pattern remains: evaluation prefers criteria with clear automatic signals, while higher-order synthesis is either coarse-grained or absent (Kasanishi et al., 2023; Su et al., 2025; Li et al., 2025). Recent work has also proposed benchmarks that aim to evaluate the academic value of survey-like outputs, e.g., DeepSurveyBench (Zhang et al., 2026).

A common response is to use LLM-as-a-judge to score richer rubrics(Hashemi et al., 2024). However, the judge model is not an oracle: its preferences can drift with prompt wording, it can overweight surface features, and it may fail precisely where domain expertise matters most. Recent meta-evaluation results on scientific, literature-grounded tasks show that even strong models only partially match expert preferences, underscoring the difficulty of using off-the-shelf judging setups as the primary development signal (Zhao et al., 2025).

LitReviewBench is designed to complement these benchmarks rather than replace them. We keep the citation-aware dimensions (coverage; citation–claim support) because they are necessary, but we center evaluation on two expert-facing dimensions that are routinely under-measured: review landscape structure and gap/direction quality. Importantly, our labels come from topic-matched researchers, and our offline evaluator(LitJudge) is calibrated to preserve those expert tradeoffs rather than optimizing for generic judge agreement.

## 2.3. Arena-Style Human Preference Evaluation

Arena-style evaluation provides an appealing alternative to static benchmarks: it supports open-ended queries, produces relative judgments that are easier to make consistently than absolute scores, and naturally yields leaderboards. Chatbot Arena (LMArena) established the now-standard recipe of blind pairwise comparison with Elo-style aggregation (Chiang et al., 2024). SciArena adapts this paradigm to non-verifiable, literature-grounded scientific tasks, explicitly targeting domains where expertise and retrieval quality are central (Zhao et al., 2025).

However, live arenas are not immediately usable as benchmarks for new methods. They are expensive to scale with experts; their query distribution evolves over time; and reproducing results for offline testing is non-trivial. LitReview Bench takes the arena advantages, topic diversity, and pairwise expert preference, but converts them into a frozen, versioned dataset with standardized per-dimension outcomes, so that progress can be measured directly offline. In addition, we introduce an expert-aligned evaluator that uses structure- and topic-matched in-context cases plus human-written gap anchors and produces quality signals, making expert-grounded evaluation and scalable training practical at low ongoing human cost.

## 3. The LitReviewBench Construction

LitReviewBench is constructed in three stages: (1) defining a survey draft generation task from high quality survey literature and mapping each instance into a consistent topic taxonomy, (2) collecting blind expert preferences via pairwise comparisons on LitReview Arena, which is part of the ScienceArena Community (Shao et al., 2025), and (3) freezing the resulting arena logs into a versioned offline benchmark that supports direct testing and standardized leaderboards.

### 3.1. Task Setup and Topic Taxonomy

LitReviewBench evaluates literature review systems on producing survey drafts that are useful to researchers. The task mirrors research practice: a good draft must not only list relevant papers but also organize a field into an interpretable landscape and surface gaps that plausibly inform next steps.

**Source pool from OpenAlex.** We retrieve papers from OpenAlex (Priem et al., 2022) with concept *Artificial Intelligence*, publication time in 2022 to 2025, and citation count $> 50$, retaining survey style papers as the seed pool (3,000+ papers). This anchors our topics in recognized research questions while staying recent enough to reflect current subfields.

**One topic extraction per survey.** From each selected survey, we use an LLM based extractor to produce a single topic phrase that captures the scope at the level of a survey prompt. We normalize these extracted topics into a consistent query form to reduce unintended variation, using the

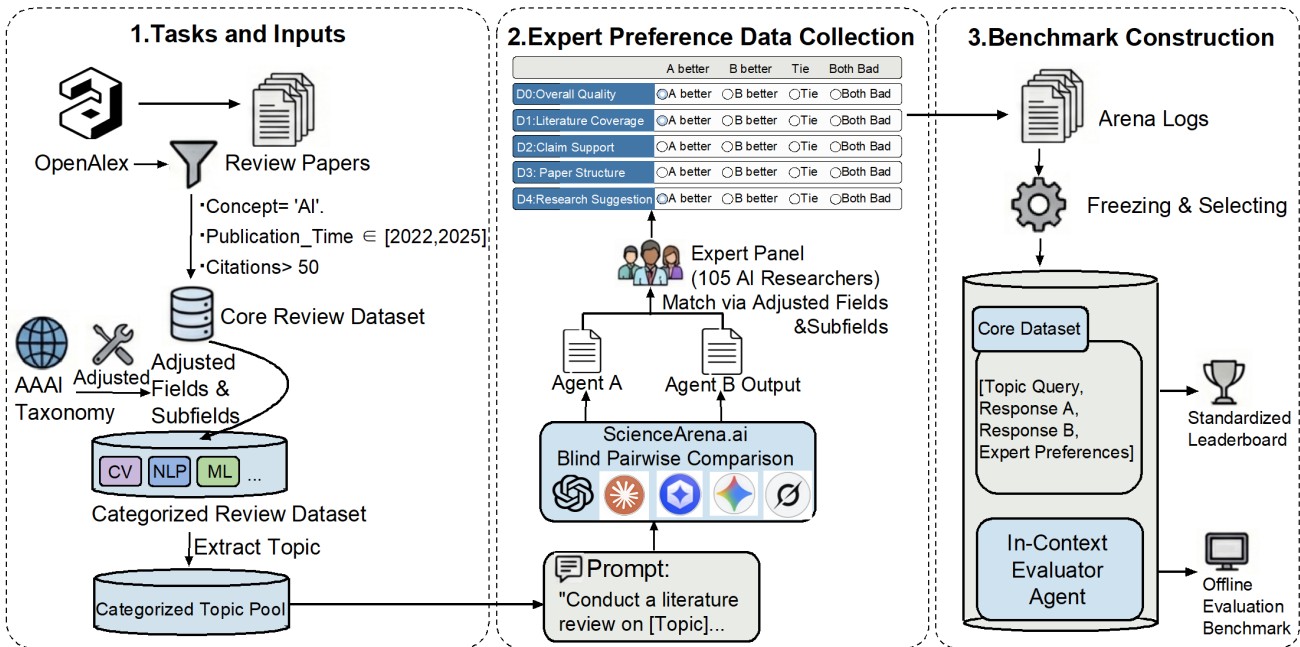

*Figure 1.* **LitReviewBench construction overview.** Left: topic extraction from high quality AI survey papers curated from OpenAlex and assignment into an AAAI based field and subfield taxonomy. Middle: expert preference collection on LitReview Arena using blind pairwise comparisons and five dimension wise four way outcomes (D1 to D5). Right: benchmark construction by freezing arena logs into a versioned dataset for offline evaluation and standardized leaderboards.

template: *Conduct a literature review on {Topic}.* Normalization focuses on de duplication and phrasing edits that preserve scope while improving clarity.

**Field and subfield assignment.** Each topic is assigned a field and subfield using an AAAI based taxonomy refined for modern AI. These tags enable expertise matching during annotation and support stratified analysis to test generalization beyond dominant areas.

### 3.2. LitReview Arena Pairwise Evaluation Protocol

LitReviewBench collects judgments through LitReview Arena, leveraging the stability of relative comparisons for open ended generation (Chiang et al., 2024; Zhao et al., 2025). For each topic, two candidate drafts generated by different systems are displayed side by side with identities hidden and order randomized.

The atomic unit is a *battle record* containing the topic query, paired drafts, and expert outcomes. This format supports standard pairwise aggregation methods used in arenas, including Bradley Terry models and Elo style ratings (Bradley & Terry, 1952; Elo, 1978). LitReviewBench preserves this battle format in the released dataset to ensure downstream evaluation follows a consistent interface.

### 3.3. Expert Preference Data Collection

We recruit 105 annotators with AI paper writing experience, matched to topics within their areas of familiarity using the field and subfield tags from Section 3.1. Annotators first complete a background questionnaire covering research areas, subfields, prior publications, and familiar papers; we then screen the responses and assign each battle to experts whose background matches the topic. This matching is critical for dimensions relying on disciplinary norms, particularly review structure and gap quality.

Before annotation, experts receive a shared instruction document that defines the five dimensions, the four outcomes (A, B, Tie, and Both Bad), and the platform rules. The interface presents anonymized side-by-side drafts with randomized left–right order, and annotators provide dimension-wise choices together with written reasoning. The guidelines also include representative positive and negative examples and warnings against common failure modes, such as overweighting surface fluency, treating citation count as sufficient coverage, or accepting generic future-work language as strong research suggestions. Each battle is evaluated by multiple matched experts when available, and we use majority judgment and expert–expert agreement checks to reduce the influence of individual preference.

Each comparison uses a four way outcome set: **A**, **B**, **Tie**, and **Both Bad**. Every battle receives five separate votes:

- **D1 Literature Coverage:** Which draft cites a more complete set of relevant papers with fewer omissions?

- **D2 Claim Support:** Which draft more reliably supports key claims with relevant citations?

- **D3 Paper Structure:** Which draft better organizes prior work into categories or comparisons that clarify relationships among approaches?

- **D4 Research Suggestions Quality:** Which draft more clearly identifies important, non obvious gaps or future directions?

- **D5 Overall Utility:** From the perspective of a researcher, which draft is preferred as a starting point?

This design yields an overall preference signal (D5) alongside four diagnostic signals (D1 to D4) that isolate aspects of scholarly synthesis where automated setups often drift. Records store minimal metadata for auditing, including pseudonymized annotator identifiers and field tags, allowing for granular analysis without embedding personal information.

### 3.4. From Arena Logs to a Reproducible Offline Benchmark

We freeze LitReview Arena logs by selecting a stable set of topics and associated battles, packaging expert outcomes as fixed labels. The resulting dataset is released in versioned form to ensure reproducibility over time.

Each benchmark instance contains the topic query, paired drafts, four way outcomes for D1 to D5, and aggregation metadata. Preserving Tie and Both Bad as explicit outcomes enables downstream evaluators to handle neutrality and quality failure separately. Finally, we support standardized leaderboards by applying the same BT and Elo style aggregation used in arena settings to the frozen outcomes (Bradley & Terry, 1952; Elo, 1978), producing comparable system ratings per dimension independent of specific judge models.

## 4. Results on SOTA Models and Agents

### 4.1. Overall Performance and Leaderboards

LitReviewBench comprises approximately 3k expert judgments, each containing five dimension-wise outcomes. We aggregate outcomes into dimension-wise leaderboards using a standard Bradley–Terry model with Elo-style ratings (Bradley & Terry, 1952; Elo, 1978). Table 1 reports the ratings for the four diagnostic dimensions (D1–D4) and Overall Utility (D5).

When competing directly against human drafts (the upper-bound reference), state-of-the-art models and agents win

only 208 out of 904 decisive matches on D5, a win rate of 23.0%.[1] This gap confirms qualitative expert feedback: functional research starting points require more than well-formed summaries.

The human baseline ranks first across all five dimensions. Among non-human systems, GPT-5.2 leads on D1–D3, while Sonar Deep Research leads on D4 (Research Suggestions), though still trailing humans. These standings indicate that current systems struggle with field organization and actionable direction-setting.

High performance correlates with computational cost. As detailed in Table 1, agentic systems consume 122.3K tokens per query on average—15× the budget of standalone models (8.1K). While allocating computation to search and synthesis yields comprehensive gains over raw models, it has not yet closed the utility gap to expert expectations.

As an additional analysis, we evaluate two systems designed specifically for literature review generation, Open Deep Research (LangChain AI, 2025) and SurveyForge (Yan et al., 2025). They substantially outperform the average agentic and language-model baselines, but still remain below human drafts across all five dimensions (Appendix A.7).

### 4.2. Fine-Grained Analysis of Model Capability Gaps

Inspection of battle records identifies four persistent failure modes: incomplete or poor literature coverage (D1), weak claim support (D2), shallow organization (D3), and generic or impractical research suggestions, such as research gaps and future directions (D4). Data from the performance gap between the top overall model, GPT-5.2, and human experts reveal that, across diagnostic dimensions D1–D4, systems consistently lag behind expert drafts, with the most pronounced deficits in Paper Structure (D3) and Research Suggestions (D4). These two dimensions are the strongest predictors of expert preference: Spearman correlations with Overall Utility (D5) are 0.99 (D3) and 0.96 (D4), exceeding that of Literature Coverage (0.90). Expert utility thus depends less on citation volume than on conceptual coherence and non-trivial future directions. Qualitatively, drafts often resemble efficient bibliographies rather than synthetic reviews.

Increasing draft length or interaction turns does not automatically yield higher D3/D4 scores. Experts reward drafts that make latent relationships legible, not those that simply expand coverage. LitReviewBench therefore exposes a reasoning gap invisible to checklist evaluations. Since D3 and D4 are also where off-the-shelf LLM judges diverge most from experts (Section 5), reliance on uncalibrated automated metrics risks misguiding development.

---

[1]Win rate calculated on decisive outcomes only.

| Category | Methods | Token Cost/Query | Literature Coverage | Claim Support | Paper Structure | Research Suggestions | Overall Utility |
|---|---|---|---|---|---|---|---|
| **Human** | **human** | **N/A** | **1787.4** | **1565.6** | **1502.5** | **1521.5** | **1668.8** |
| **Agentic Models** | GPT-5.2 | 38.096K | 1632.8 | 1536.5 | 1322.4 | 1272.7 | 1449.1 |
| | Sonar Deep Research | 322.080K | 1175.9 | 1106.3 | 1262.1 | 1322.7 | 1285.9 |
| | Qwen Deep Research | 70.265K | 863.3 | 925.4 | 1125.0 | 1178.4 | 1117.3 |
| | OpenAI Deep Research | 58.745K | 882.2 | 943.3 | 857.5 | 867.3 | 874.1 |
| | **Average** | **122.297K** | **1138.6** | **1127.9** | **1141.8** | **1160.3** | **1181.6** |
| **Language Models** | Claude Opus 4.5 | 5.490K | 1129.0 | 1032.8 | 1177.6 | 1099.9 | 1135.5 |
| | Qwen3 235B | 5.276K | 759.3 | 826.3 | 886.0 | 877.4 | 836.2 |
| | Grok 4 | 16.399K | 886.8 | 908.5 | 780.2 | 778.2 | 799.2 |
| | GLM 4.6 | 4.741K | 436.7 | 553.6 | 564.0 | 608.5 | 434.3 |
| | Gemini 2.5 Pro | 8.586K | 446.1 | 601.8 | 526.0 | 469.8 | 400.2 |
| | **Average** | **8.098K** | **731.6** | **784.6** | **786.8** | **766.8** | **721.1** |

*Table 1.* Expert-preference leaderboards on LitReviewBench. *Token Cost/Query* (third column): lower is better, reported in thousands of tokens based on exact measurements. *Utility metrics* (columns 4 to 8): higher is better, ordered as Literature Coverage, Claim Support, Paper Structure, Research Suggestions, and Overall Utility. Systems are grouped as Human, Agentic Models, and Language Models. Human achieves the highest utility scores across all metrics; GPT-5.2 leads non-human systems in Overall Utility.

These findings also clarify why benchmark design matters for literature review agents. If evaluation focuses mainly on reference overlap, citation count, or broad topical coverage, systems can appear strong while still failing to produce the organizing arguments that make a review useful. The dimension-wise arena format makes this failure visible: a system can retrieve relevant work and still lose on Paper Structure or Research Suggestions when experts judge that the draft does not explain how subareas relate or what unresolved questions matter. This separation between coverage and synthesis is central to LitReviewBench's diagnostic value.

## 5. Meta-Evaluation of LLM-Based Judges

Collecting expert preferences is resource-intensive, raising the question of whether LLM-based judges serve as reliable proxies. Prior work suggests scientific literature tasks pose greater challenges than general chat evaluation, with evaluators often failing to match expert consensus (Zhao et al., 2025). We conduct a meta-evaluation on LitReviewBench to quantify this alignment gap.

**Setup.** We sample 500 battle instances from LitReviewBench, preserving the task format and 4-way expert votes. We employ Qwen/Qwen3-235B-A22B-Instruct-2507 as the automated judge using a minimal prompt that forces a decision for each of the five dimensions. We run the judge once per instance and dimension.

**Scoring and aggregation.** We evaluate alignment via instance-level accuracy, assigning 0.5 credit for neutral expert outcomes, and leaderboard-level Spearman correlation. Judge-induced system rankings are derived using the same Bradley–Terry aggregation as Section 4.1 to enable direct

comparison with expert standings (Table 2). Table 3 summarizes agreement and reliability.

### 5.1. Blind Spots of LLM Judges: Systematic Mismatches with Human Experts

We observe that the judge's reliability varies fundamentally depending on the nature of the evaluation sub-task. On Literature Coverage (D1), which primarily tests retrieval recall, Qwen correlates moderately well with experts ($\rho = 0.552$). The model effectively identifies whether a draft includes the expected set of canonical papers.

However, performance degrades as the task shifts from retrieval to synthesis. On Claim Support (D2), Paper Structure (D3), and Research Suggestions (D4), the uncalibrated judge struggles to differentiate high-quality analysis from plausible but incorrect statements.

More critically, Table 2 reveals a severe AI to AI bias. While experts rank human drafts first (Table 1), the automated judge penalizes human writing aggressively. On Overall Utility (D5), it assigns humans an Elo score of 310 while boosting GPT-5.2 to 2490. This inversion suggests the judge conflates model-like fluency with quality, making it unreliable for comparing human and machine outputs without calibration.

### 5.2. Consistency and Reliability

A common hypothesis attributes disagreement on high-level dimensions to inherent task subjectivity. We test this by comparing agreement accuracy for expert pairs versus judge pairs. For judge pairs, we measure cross-model agreement between Qwen/Qwen3-235B-A22B-Instruct-2507 and DeepSeek-V3.2 to assess whether failure patterns remain consistent across different LLM judges.

| Category | Methods | Literature Coverage | Claim Support | Paper Structure | Research Suggestions | Overall Utility |
|---|---|---|---|---|---|---|
| Human | **human** | **378** | **321** | **317** | **407** | **310** |
| **Agentic Models** | GPT-5.2 | 2439 | 2470 | 2485 | 2367 | 2490 |
| | Sonar Deep Research | 2012 | 2071 | 2094 | 1864 | 2096 |
| | Qwen Deep Research | 1320 | 1320 | 1371 | 1392 | 1370 |
| | OpenAI Deep Research | 802 | 767 | 763 | 775 | 761 |
| | **Average** | **1643.3** | **1657.0** | **1678.3** | **1599.5** | **1679.3** |
| **Language Models** | Claude Opus 4.5 | 1163 | 1033 | 1035 | 915 | 1068 |
| | Qwen3 235B | 848 | 926 | 945 | 1069 | 923 |
| | Grok 4 | 575 | 559 | 524 | 625 | 521 |
| | GLM 4.6 | 293 | 305 | 255 | 350 | 251 |
| | Gemini 2.5 Pro | 129 | 174 | 165 | 203 | 165 |
| | **Average** | **601.6** | **599.4** | **584.8** | **632.4** | **585.6** |

*Table 2.* Judge-induced leaderboards on LitReviewBench using Qwen/Qwen3-235B-A22B-Instruct-2507 as the evaluator. Scores are Elo ratings derived from Bradley–Terry aggregation, and higher is better.

| Dimension | Judge–Expert Accuracy | Spearman's $\rho$ | Expert–Expert Agreement Accuracy | JudgeLLM–JudgeLLM Agreement Accuracy |
|---|---|---|---|---|
| D1 (Literature Coverage) | 0.586 | **0.552** | 0.833 | **0.783** |
| D2 (Claim Support) | 0.554 | 0.442 | 0.556 | 0.747 |
| D3 (Paper Structure) | 0.598 | 0.467 | 0.639 | 0.747 |
| D4 (Research Suggestions) | **0.620** | 0.430 | 0.556 | 0.739 |
| D5 (Overall Utility) | 0.606 | 0.467 | **0.861** | 0.747 |

*Table 3.* Agreement with expert preference and reliability. Judge–expert accuracy merges Tie and Both Bad into a neutral outcome and assigns 0.5 credit regardless of the judge decision. Spearman's $\rho$ is the rank correlation between judge-induced and expert-induced BT and Elo leaderboards. Expert–expert agreement accuracy and JudgeLLM–JudgeLLM agreement accuracy are reported as mean pairwise accuracy. JudgeLLM–JudgeLLM agreement measures cross-model consistency between two judge models, Qwen/Qwen3-235B-A22B-Instruct-2507 and DeepSeek-V3.2, indicating whether failure patterns remain stable across different LLM judges.

Expert agreement is consistently higher than judge agreement on D3 and D4, reaching 0.861 on Overall Utility (D5) (Table 3). This high consensus indicates that the quality signal is not dominated by noise. Notably, judge-to-judge agreement remains high ($> 0.7$) across all dimensions, even where alignment with experts is poor. We therefore interpret high LLM–LLM agreement on D2 and D4 not as evidence of expert-level reliability, but as evidence of shared AI-side preferences. LLM judges may converge on surface fluency, formatting quality, or a common evaluation style while under-weighting whether claims are genuinely grounded in cited literature or whether proposed research directions are insightful and non-obvious. This indicates that judge failures are systematic rather than random, reflecting shared biases toward surface polish over deep structure.

### 5.3. Implications for Evaluator Calibration

Naive LLM judging is an insufficient substitute for expert preference. The capabilities required to assess claim grounding, structure, and research gaps differ significantly from those required for simple coverage checking. Section 6.2 further reports the same naive-versus-calibrated comparison across GPT, Claude, and Llama judge backbones, showing

that the gap is not specific to Qwen.

Leaderboards derived from uncalibrated judges are deceptively stable yet inaccurate. As shown in Table 2, judge-based rankings systematically undervalue human baselines and over-weight surface-level mechanics. These limitations motivate calibrating the evaluator with task-specific in-context examples, leading to our calibrated evaluator in Section 6.1.

## 6. Closing the Loop: Expert-Aligned Evaluator

Progress on literature review agents is constrained less by generation capabilities than by evaluation reliability. Standard LLM-based evaluators perform adequately on Literature Coverage (D1) but fail to capture expert utility on Claim Support (D2), Paper Structure (D3), and Research Suggestions (D4). We address this gap by using LitReviewBench as a calibration signal and instantiating an expert-aligned evaluator, LitJudge. As shown in Figure 2, LitJudge conditions on case context that matches the test instance in structure and topic, and it grounds D4 decisions with diversity-aware gap anchors derived from expert-written drafts.

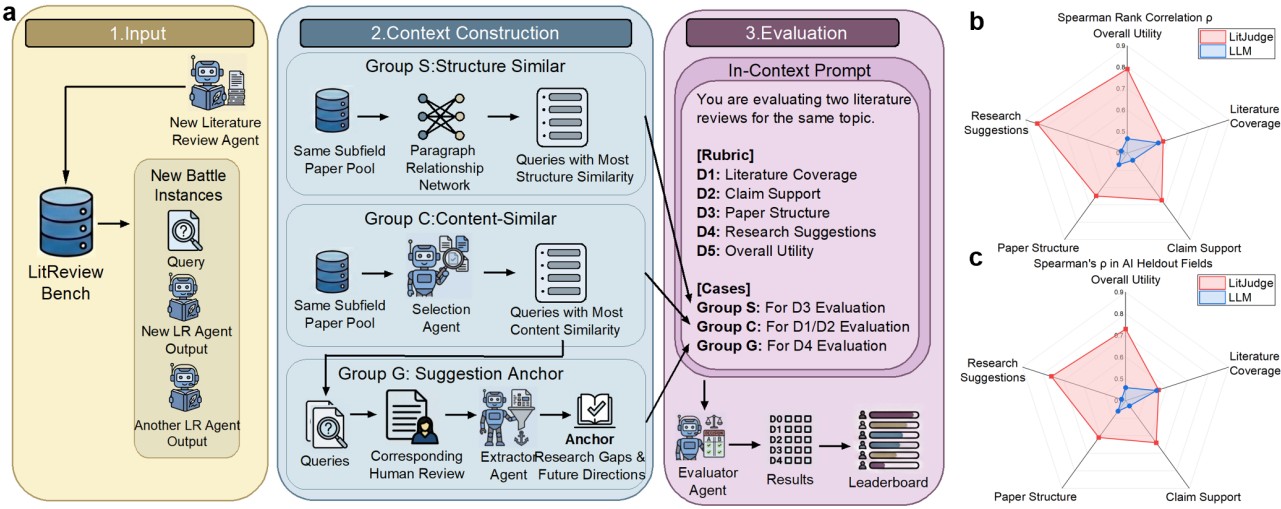

*Figure 2.* Expert-aligned evaluator workflow and results. (a) Context construction for the calibrated evaluator. (b) Calibration improves alignment between judge and expert votes. (c) In-domain generalization to held-out AI subfields (20%) shows robust transfer.

### 6.1. Expert-in-the-Loop Calibration

We employ Qwen/Qwen3-235B-A22B-Instruct-2507, consistent with Section 5, and use a 500 instance subset from LitReviewBench for calibration and evaluation.

Calibrated evaluator context (Figure 2a). For each battle, we construct an in-context packet with up to three demonstrations per group:

- Group S (Structure-similar; for D3). We extract a skeleton text (headers and lead sentences) to derive a paragraph relationship network capturing discourse transitions. We retrieve battles with the most similar networks based on normalized graph similarity.

- Group C (Content-similar; for D1/D2). We retrieve battles with topics closest to the query using LLM-based embedding matching. These cases provide local standards for sufficient coverage and plausible citation–claim support.

- Group G (Gap anchors; for D4). We extract gap anchors exclusively from expert-written human drafts. These bullet points serve as grounding exemplars for high-quality, non-hallucinated directions. To avoid concentrating the context on a narrow set of recurring suggestions, LitJudge applies maximal marginal relevance when selecting D4 anchors, balancing relevance with diversity.

### 6.2. Efficacy of the Calibrated Evaluator

We evaluate judge alignment by agreement accuracy against expert votes, using the same scoring protocol as Section 5. Figure 2b summarizes the gains from calibration.

LitJudge improves alignment most on synthesis dimensions, while keeping coverage behavior comparable. On Literature Coverage (D1), the gain is modest, from 0.552 to 0.576. On Claim Support (D2) and Paper Structure (D3), alignment increases substantially, from 0.442 to 0.673 and from 0.467 to 0.649. The largest jump is on Research Suggestions (D4), from 0.430 to 0.842, consistent with the role of expert-derived gap anchors. Overall Utility (D5) also rises sharply, from 0.467 to 0.792, indicating that calibration helps recover the expert holistic preference signal.

Appendix A.8 further shows that the gain is not tied to a specific judge backbone and is not merely a generic few-shot effect: LitJudge improves over naive judging across GPT, Claude, Llama, and Qwen backbones, and it outperforms a random few-shot ICL control using the same number of examples.

This enables repeated offline evaluation that tracks expert tradeoffs with low marginal cost.

## 7. Discussion

LitReview Arena targets a practical bottleneck for literature review agents: progress is now limited more by how we measure scholarly value than by how fluently systems can write. Our results show a clear split across dimensions. When the judgment is close to retrieval verification, such as Literature Coverage (D1), uncalibrated LLM judges track experts moderately well. When the judgment depends on synthesis, especially Paper Structure (D3) and Research Suggestions (D4), the same judges become unreliable, including large ranking inversions between human drafts and model outputs. This is consistent with expert feedback that utility is driven by organizing axes and actionable gaps, not by surface completeness alone.

The model results point to the same bottleneck. Agentic sys-

tems and specialized literature-review systems improve over base language models, suggesting that search, decomposition, and iterative synthesis are useful ingredients. However, their remaining gaps to human drafts are largest on D3 and D4, not on surface-level coverage. This indicates that simply allocating more test-time computation or retrieving more papers is insufficient: systems must learn to impose a meaningful conceptual structure on a field and to identify research directions that are specific, grounded, and non-obvious.

Methodologically, LitReview Arena turns expert peer-review preferences into a scalable and repeatable benchmark interface. Relative votes are easier to apply consistently than absolute scores, and the explicit outcomes Tie and Both Bad help separate indifference from quality failure. Freezing arena logs into LitReviewBench then makes this live preference signal reproducible for offline evaluation and standardized leaderboards, while the live arena remains a natural mechanism for future refreshes as topics, systems, and expert expectations change.

The calibrated evaluator closes part of the loop. Lit-Judge conditions the judge on task-specific context, including structure-matched cases and diversity-aware expert-derived gap anchors, making offline iteration feasible at low marginal cost. Newly emerging topics and expert feedback can refresh the benchmark, expand the expert pool, and update the topic classification system. The diversity-aware anchor selection is intended to reduce the risk of an academic echo chamber by lowering concentration on recurring research directions while preserving agreement with expert judgment.

Several limitations remain. First, the main benchmark is anchored in AI topics. A biology pilot study provides initial evidence that the same protocol can produce meaningful cross-domain expert judgments, but broader validation across fields with different evidential norms remains future work. Fields such as biomedicine or law may require stricter factual, safety, and citation audits in addition to preference judgments. Second, expert-calibrated preferences can entrench biases if structure matching favors familiar rhetorical forms or if gap anchors over-represent fashionable directions. This also creates a potential negative societal impact: if used uncritically, an evaluator calibrated to existing expert preferences may amplify mainstream academic styles or fashionable topics and discourage unconventional but valuable forms of scientific synthesis. Finally, the current benchmark does not cover long-horizon settings such as maintaining living reviews, where systems must track new papers, revise claims, and preserve consistency over time. Complementary evaluations, including targeted factual audits, citation-grounding checks, and longitudinal update tasks, are promising directions.

## 8. Conclusion

We introduced LitReview Arena, a battle-style peer review platform for literature review agents, and LitReviewBench, an offline benchmark distilled from arena logs. Analyzing 3k expert judgments, each containing five dimension-wise outcomes, we show that current SOTA systems still lag behind human drafts, particularly in organizing coherent landscapes and generating non-obvious research insights. We also show that uncalibrated LLM judges are unreliable for synthesis-heavy tasks, and introduce LitJudge, a calibrated evaluator using structure-matched cases and diversity-aware expert-derived anchors to improve agreement with experts. Beyond reporting a leaderboard, the benchmark identifies where progress is needed: future literature review agents should move from broad paper collection toward defensible synthesis, grounded claims, and field-level reasoning. Overall, LitReview Arena, LitReviewBench, and LitJudge align development signals for literature review agents with actual researcher needs.

## Impact Statement

This work aims to improve the evaluation of AI systems that generate scientific literature reviews by making assessment more transparent, reproducible, and aligned with expert judgment. The benchmark and LitJudge may help researchers identify weaknesses in coverage, claim support, structure, and research suggestions before such systems are used in research workflows. Potential risks include over-reliance on automated evaluators and reinforcement of dominant research styles if expert-calibrated preferences are used uncritically. LitJudge should therefore be used as a development and triage signal rather than a replacement for human expert review, and preference-based evaluation should be combined with factual checks of claims and citations.

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

# A. Appendix

### A.1. Expert Evaluation Protocol

**Outcome set.**   Each dimension is labeled with one of {A, B, Tie, BothBad}.

- **A**: Draft A is better on this dimension.

- **B**: Draft B is better on this dimension.

- **Tie**: both drafts are comparably good.

- **BothBad**: neither draft is acceptable.

**Dimension questions (verbatim).**

- **D5 Overall Utility:** From a researcher's perspective, which response would you prefer to use as a starting point for a literature review on this topic?

- **D1 Literature Coverage:** Which response cites a more complete and appropriate set of relevant papers for this topic, with fewer obvious omissions of important work?

- **D2 Claim Support:** Which response more reliably grounds its key claims in the cited literature, with citations that are relevant and appropriately support the statements made?

- **D3 Paper Structure:** Which response better structures the existing literature by organizing prior work into clear categories or comparisons that help a researcher understand relationships among approaches, rather than listing papers independently?

- **D4 Research Suggestions Quality:** Which response more clearly identifies important and non-obvious research gaps or future directions that go beyond generic summaries and would meaningfully inform a researcher's next steps?

**Interface rules.**   Drafts are anonymized and displayed side by side with randomized left–right order. Annotators are instructed to judge each dimension independently, provide written reasoning, and avoid relying on surface fluency alone. The instruction document explicitly warns against equating more citations with better coverage and against treating generic future-work statements as high-quality research suggestions. Two drafts are presented side-by-side with anonymized system identity and randomized left/right ordering. Annotators are allowed to consult external resources (e.g., follow citations / search for papers) when forming judgments.

### A.2. Data Schema (Battle Record)

**Fields.**   Each battle record contains:

- `query`: normalized topic prompt, formatted as `Conduct a literature review on {Topic}.`

- `response_a`, `response_b`: draft texts.

- `label_d1`...`label_d5`: {A,B,Tie,BothBad}.

- `field`, `subfield`: taxonomy tags.

- `annotator_id`: pseudonymized identifier.

- `metadata`: optional audit fields (e.g., hashed system ids).

**JSON example.**

```
{
  "query": "Conduct a literature review on ...",
  "response_a": "...",
  "response_b": "...",
  "label_d1": "Tie",
  "label_d2": "B",
  "label_d3": "A",
  "label_d4": "BothBad",
  "label_d5": "A",
  "field": "...",
  "subfield": "...",
  "annotator_id": "anon_XXXX",
  "metadata": {"...": "..."}
}
```

### A.3. Aggregation and Scoring Conventions

**Bradley–Terry (BT).** We fit a BT model per dimension with system-level parameters. Ties are treated as half wins for each side:

$$\texttt{Tie} \Rightarrow 0.5 \text{ win for A } + 0.5 \text{ win for B.}$$

**Elo.** We compute Elo per dimension with `init`=1500 and $K = 32$. Tie is treated as a 0.5 score for each side (consistent with BT).

**Judge–expert alignment (neutral credit).** For alignment accuracy, `Tie` and `BothBad` are treated as neutral expert outcomes. If the expert label is `Tie` or `BothBad`, the judge receives 0.5 credit regardless of output `A` or `B`.

### A.4. Meta-Evaluation Judge

**Judge model.** Qwen/Qwen3-235B-A22B-Instruct-2507.

**Prompt.**

```
{"role": "system",
 "content": "You are an expert evaluator for literature reviews. You must provide
    judgments as JSON."}

{"role": "user", "content": """
You are an expert evaluator for literature reviews. Your task is to compare two draft
    literature reviews
(Draft A and Draft B) and make judgments across five dimensions.

Evaluation Dimensions:
D5: Overall Utility  From a researcher's perspective, which response would you prefer to
    use as a starting point
for a literature review on this topic?
D1: Literature Coverage  Which response cites a more complete and appropriate set of
    relevant papers for this topic,
with fewer obvious omissions of important work?
D2: Claim Support  Which response more reliably grounds its key claims in the cited
    literature, with citations that are
relevant and appropriately support the statements made?
D3: Paper Structure  Which response better structures the existing literature by
    organizing prior work into clear
categories or comparisons that help a researcher understand relationships among
    approaches, rather than listing papers
independently?
```

```
D4: Research Suggestions Quality  Which response more clearly identifies important and
   non-obvious research gaps or
future directions that go beyond generic summaries and would meaningfully inform a
   researcher's next steps?

For each dimension, you must choose one of: A, B, Tie, or BothBad.
- A: Draft A is better
- B: Draft B is better
- Tie: Both drafts are equally good
- BothBad: Neither draft is acceptable
```

## A.5. Calibrated Expert-Aligned Evaluator

**Group sizes.**  $k_S = 3$ (structure-similar examples), $k_C = 3$ (content-similar examples), $k_G = 3$ (gap anchors).

```
{"role": "system",
 "content": "You are an expert evaluator for literature reviews. You must provide
   judgments as JSON."}

{"role": "user", "content": """
You are an expert evaluator for literature reviews. Your task is to compare two draft
   literature reviews
(Draft A and Draft B) and make judgments across five dimensions.

Evaluation Dimensions:
D5: Overall Utility  From a researcher's perspective, which response would you prefer to
   use as a starting point
for a literature review on this topic?
D1: Literature Coverage  Which response cites a more complete and appropriate set of
   relevant papers for this topic,
with fewer obvious omissions of important work?
D2: Claim Support  Which response more reliably grounds its key claims in the cited
   literature, with citations that are
relevant and appropriately support the statements made?
D3: Paper Structure  Which response better structures the existing literature by
   organizing prior work into clear
categories or comparisons that help a researcher understand relationships among
   approaches, rather than listing papers
independently?
D4: Research Suggestions Quality  Which response more clearly identifies important and
   non-obvious research gaps or
future directions that go beyond generic summaries and would meaningfully inform a
   researcher's next steps?

For each dimension, you must choose one of: A, B, Tie, or BothBad.
- A: Draft A is better
- B: Draft B is better
- Tie: Both drafts are equally good
- BothBad: Neither draft is acceptable

=== Structure-Similar Examples (for D3) ===
Example 1:
Query: [STRUCT_EX_QUERY_1]
Draft A:
[STRUCT_EX_A_1]
Draft B:
[STRUCT_EX_B_1]
Expert Outcomes:
  D1: [STRUCT_EX_D1_1]
```

```
  D2: [STRUCT_EX_D2_1]
  D3: [STRUCT_EX_D3_1]
  D4: [STRUCT_EX_D4_1]
  D5: [STRUCT_EX_D5_1]
[... up to k_S=3 ...]

=== Content-Similar Examples (for D1/D2) ===
Example 1:
Query: [CONT_EX_QUERY_1]
Draft A:
[CONT_EX_A_1]
Draft B:
[CONT_EX_B_1]
Expert Outcomes:
  D1: [CONT_EX_D1_1]
  D2: [CONT_EX_D2_1]
  D3: [CONT_EX_D3_1]
  D4: [CONT_EX_D4_1]
  D5: [CONT_EX_D5_1]
[... up to k_C=3 ...]

=== Gap Anchors (for D4) ===
Examples of meaningful research gaps and future directions from expert-written reviews:
1. [GAP_ANCHOR_1]
2. [GAP_ANCHOR_2]
3. [GAP_ANCHOR_3]

=== Current Battle to Evaluate ===
Query: [CUR_QUERY]
Draft A:
[CUR_DRAFT_A]
Draft B:
[CUR_DRAFT_B]

Return a JSON object:
{
  "D1": "A|B|Tie|BothBad",
  "D2": "A|B|Tie|BothBad",
  "D3": "A|B|Tie|BothBad",
  "D4": "A|B|Tie|BothBad",
  "D5": "A|B|Tie|BothBad"
}
Do not output any other keys or values.
"""}
```

**Output schema.**   A single JSON object with keys `D1`–`D5` and values in {A,B,Tie,BothBad}.

### A.6. Additional Human Evaluation Cases

**Case 1: validation metrics in medical image analysis.**   The query asks for a literature review on validation metrics in medical image analysis. Draft A reviews metrics across segmentation, classification, and detection, and discusses class imbalance, calibration, sensitivity–specificity trade-offs, clinically meaningful evaluation, and deployment risks. It covers representative metrics broadly and explains why metric choice affects interpretation and downstream use, but its organization is more sequential. Draft B covers a narrower set of metrics and settings, but structures the discussion around clearer evaluation axes, including overlap-based metrics, ranking-based metrics, calibration-oriented metrics, and task-specific clinical relevance.

The expert outcomes are: Literature Coverage = A, Claim Support = A, Paper Structure = B, Research Suggestions = A, and Overall Utility = A. The written explanation is that Draft A is broader and more useful as a starting point because it goes beyond listing metric families and connects metric choice to practical clinical risks. Draft B is preferred only on Paper

Structure because it organizes the literature around clearer axes.

**Case 2: dopaminergic reward system.** The query asks for a literature review on the organization and function of the dopaminergic reward system. Draft A, generated by Claude Opus 4.5, is organized around the ventral tegmental area, dopamine-neuron heterogeneity, reward prediction error signaling, the relationship between firing and dopamine release, and clinical implications. Draft B, generated by Grok 4, is organized around the mesolimbic, mesocortical, and nigrostriatal pathways, receptor-level distinctions, synaptic plasticity, stress-related modulation, and implications for addiction and other disorders.

The expert outcomes are: Literature Coverage = B, Claim Support = B, Paper Structure = A, Research Suggestions = B, and Overall Utility = B. The annotator explanation for Literature Coverage was: "An important fact to understand about the dopaminergic circuit is that there is currently no unified explanation of dopaminergic circuit, and it serves complex circuit-specific function. This was not explained in A."

### A.7. Specialized Literature Review Systems

We additionally evaluate systems specifically designed for scientific writing or literature review generation on a subset of LitReviewBench.

| Method | D1 | D2 | D3 | D4 | D5 |
|---|---|---|---|---|---|
| Human | 1979.8 | 1955.8 | 1880.4 | 1871.7 | 1912.2 |
| Open Deep Research (LangChain AI, 2025) | 1845.6 | 1812.2 | 1868.5 | 1811.0 | 1832.5 |
| SurveyForge (Yan et al., 2025) | 1799.1 | 1775.5 | 1842.7 | 1797.2 | 1805.7 |
| Agentic Models Avg. | 1485.5 | 1468.6 | 1478.2 | 1537.2 | 1511.5 |
| LLMs Avg. | 1250.8 | 1280.5 | 1259.8 | 1255.6 | 1248.4 |

*Table 4.* Specialized literature review systems outperform average model baselines but remain below human drafts.

The specialized systems substantially improve over both average agentic and language-model baselines across all five dimensions. Open Deep Research obtains the strongest specialized-system results, with especially small gaps to human drafts on Paper Structure (D3), while SurveyForge shows a similar pattern at slightly lower ratings. However, both systems remain below human drafts on every dimension, indicating that systems explicitly designed for literature review generation narrow but do not close the expert-preference gap.

## A.8. Additional LitJudge Analyses

Table 5 reports two additional controls for LitJudge. First, the naive-versus-calibrated pattern holds across GPT, Claude, Llama, and Qwen backbones, showing that the effect is not specific to one evaluator model. Second, a random few-shot ICL judge using the same number of examples remains below LitJudge on Claim Support, Paper Structure, Research Suggestions, and Overall Utility, indicating that the gains come from task-specific retrieval and diversity-aware calibration rather than adding examples alone.

| Base Model / Method | Setting | D1 | D2 | D3 | D4 | D5 |
|---|---|---|---|---|---|---|
| Qwen3-235B | Naive Judge | 0.552 | 0.442 | 0.467 | 0.430 | 0.467 |
|  | LitJudge | 0.576 | 0.673 | 0.649 | 0.842 | 0.792 |
| GPT-5.4 | Naive Judge | 0.673 | 0.370 | 0.721 | 0.766 | 0.770 |
|  | LitJudge | 0.855 | 0.867 | 0.830 | 0.879 | 0.952 |
| Claude-Sonnet-4.5 | Naive Judge | 0.779 | 0.609 | 0.704 | 0.758 | 0.809 |
|  | LitJudge | 0.818 | 0.855 | 0.842 | 0.900 | 0.976 |
| Llama-4-maverick | Naive Judge | 0.588 | 0.539 | 0.576 | 0.709 | 0.636 |
|  | LitJudge | 0.594 | 0.891 | 0.806 | 0.842 | 0.673 |
| Qwen3-235B | Random few-shot ICL | 0.484 | 0.554 | 0.583 | 0.737 | 0.634 |
| Qwen3-235B | LitJudge | 0.576 | 0.673 | 0.649 | 0.842 | 0.792 |

*Table 5.* Spearman's $\rho$ between judge-induced and expert-induced leaderboards. LitJudge improves over naive judging across multiple base models and outperforms a random few-shot ICL control using the same number of examples, indicating that gains come from task-specific retrieval and diversity-aware calibration rather than few-shot prompting alone.

## A.9. Biology Pilot Study

We conduct a pilot cross-domain analysis in biology using the same arena-style expert annotation protocol. The results follow the same high-level pattern as the AI benchmark: human drafts outperform agentic models, which in turn outperform language models.

| Category | D1 | D2 | D3 | D4 | D5 |
|---|---|---|---|---|---|
| Human | 1947.70 | 1742.00 | 2000.61 | 2007.63 | 2267.27 |
| Agentic Models Avg. | 1701.97 | 1612.13 | 1465.10 | 1522.75 | 1593.52 |
| Language Models Avg. | 1289.28 | 1384.32 | 1420.82 | 1384.82 | 1290.43 |

*Table 6.* Biology pilot leaderboard. Agentic models include OpenAI Deep Research, Qwen Deep Research, and GPT-5.2. Language models include Claude Opus 4.5, Qwen3 235B, Grok 4, GLM 4.6, and Gemini 2.5 Pro.

The largest human-model gaps still appear on Paper Structure and Research Suggestions. Relative to Agentic Models, the gap to human experts is 535.51 on D3 and 484.88 on D4, compared with 245.73 on D1 and 129.87 on D2. This pattern is even more pronounced than in the AI-domain results in the main paper. A plausible explanation is that biology reviews often require the organization of heterogeneous evidence across multiple levels, such as molecules, cells, circuits, and phenotypes, while also requiring careful treatment of unresolved mechanisms and competing interpretations. These properties place greater demands on synthesis, field organization, and scientific judgment, which makes D3 and D4 particularly challenging for current models.

| Judge | D1 | D2 | D3 | D4 | D5 |
|---|---|---|---|---|---|
| Naive Judge | 0.7367 | 0.5467 | 0.4700 | 0.5133 | 0.5900 |
| LitJudge | 0.8833 | 0.6167 | 0.6000 | 0.7333 | 0.8833 |

*Table 7.* LitJudge improves over the naive judge on the biology pilot study.

## A.10. Annotator Survey (Background) and Summary Statistics

**Questionnaire items.** Tencent questionnaire items: Q01 Name; Q02 pre-survey completion; Q03 phone number; Q04 NLP; Q05 CV; Q06 ML; Q07 Reasoning & Symbolic AI; Q08 Data Mining & Big Data; Q09 Robotics & Embodied AI;

Q10 Multi-Agent & Game Theory; Q11 Interdisciplinary Applications; Q12 other directions; Q13 prior publications (DOIs); Q14 most familiar papers (at least 5 DOIs).

**Survey statistics (from responses).** Number of respondents: $N = 107$. Completion time (seconds): median $= 510$, IQR $= [293, 990]$, min $= 31$, max $= 107443$.

**Top-level area coverage.** Count of respondents selecting at least one subtopic: NLP (91), CV (71), ML (74), Reasoning & Symbolic AI (60), Data Mining & Big Data (64), Robotics & Embodied AI (50), Multi-Agent & Game Theory (66), Interdisciplinary Applications (76).

### A.11. Field/Subfield Taxonomy (Full List)

The field/subfield taxonomy is directly instantiated from the annotator background questionnaire (Tencent Survey, Q04–Q11). Each field corresponds to one multi-select question, and each subfield corresponds to a selectable option.

**Q04: Natural Language Processing (NLP).**

- Large Language Models (LLMs)

- Prompt Engineering and In-Context Learning

- Text Generation and Summarization

- Conversational AI and Dialogue Systems

- Machine Translation and Multilingual NLP

- Information Extraction and Knowledge-related NLP

- NLP Evaluation, Analysis, and Interpretability

- NLP Safety, Bias, and Fact-checking

**Q05: Computer Vision (CV).**

- Generative Vision Models

- Large Vision Models and Foundation Models

- Vision–Language and Multimodal Learning

- Image and Video Understanding

- 3D Computer Vision

- Medical and Biological Imaging

- Low-level Vision and Computational Photography

**Q06: Machine Learning (Methodologies).**

- Deep Learning Architectures

- Reinforcement Learning

- Graph Machine Learning

- Trustworthy and Robust Machine Learning

- Optimization and Learning Theory

- Self-supervised and Unsupervised Learning

- Federated and Distributed Learning

- Neuro-Symbolic AI

**Q07: Reasoning, Planning, and Symbolic AI.**

- Knowledge Representation and Reasoning

- Knowledge Graphs

- Automated Planning and Scheduling

- Search, Optimization, and Constraint Satisfaction

- Causality

- Reasoning under Uncertainty

**Q08: Data Mining and Big Data.**

- Recommender Systems

- Time-series Analysis

- Anomaly and Outlier Detection

- Web Mining and Social Computing

- Spatio-temporal Data Mining

- Databases and Data Management for AI

**Q09: Robotics and Embodied AI.**

- Embodied AI

- Robot Learning

- SLAM and Navigation

- Multi-robot Systems

- Human–Robot Interaction (HRI)

**Q10: Multi-Agent Systems and Game Theory.**

- Multi-agent Coordination and Collaboration

- Game Theory and Economic Paradigms

- Agent-based Modeling and Simulation

- Social Choice and Voting

**Q11: Interdisciplinary Applications and Society.**

- AI for Science

- Cognitive Modeling and Cognitive Systems

- Human–AI Collaboration and HCI

- AI Ethics, Law, and Governance

- Smart Cities and Transportation

- Financial Technology (FinTech)

