# OpenReview forum: "LitReview Arena: Evaluating Literature Review Agents with Battle-style Peer Review Platform"
_ICML.cc/2026/Conference — ICML 2026 regular_

### Official Review · Reviewer_Wzs8 · 2026-03-10

**Soundness:** 3
**Presentation:** 3
**Significance:** 3
**Originality:** 3
**Overall Recommendation:** 4
**Confidence:** 3

**Summary:**

This paper introduces LitReview Arena, a battle-style peer review platform designed to evaluate the quality of literature reviews. The papoer build LitReviewBench, collecting 3K expert annotation across five dimensions with AI researchers with expertise in their respective domains. They evaluate human review and AI-generated review with pairwise comparison, showing the human outperform AI in large performance gap, even strongest models(Agentic models). Furthermore, they demonstrate the LLM-as-a-judge is misaligned with expert preference, and propose LitJuge, an in-context learning-based evaluator that improves alignment from 0.467 to 0.78.

**Compliance With Llm Reviewing Policy:**

Affirmed.

**Final Justification:**

This paper addresses a timely and important problem: the lack of reliable evaluation frameworks for LLM-based literature review generation. The core contribution -- constructing LitReviewBench with expert-grounded pairwise preference data across five structured dimensions -- is meaningful and costly to produce, and I acknowledge its significance.
The rebuttal addressed several concerns satisfactorily. However, two concerns remain unresolved. Overall, I view the benchmark construction as the primary contribution of this work, and I have updated my score accordingly.

**Key Questions For Authors:**

- Have you tried other model-series except Qwen? such as GPT, Claude, Llama, or else for LLM evaluator?
- In the discussion, the authors acknowledge "pairwise comparisons do not fully disentangle writing style from substance". Then, how do you ensure the objectivity of the benchmark? Were any controls applied during annotation to mitigate this confound or does it remain for future work?
- The paper describes five evaluation dimensions. However, the annotation guidelines (Appendix A.1) do not include examples to articulate the dimension. And what was the instruction for annotators beyond what is listed in Appendix A.1?
- LitJudge is calibrated using in-context examples drawn from LitReviewBench and evaluated on a held-out split of the same benchmark. Can you provide evidence of generalization to an independently constructed preference dataset?

**Limitations:**

yes

**Strengths And Weaknesses:**

> Strenght

- The paper proposed timely and well-motivated problems.
- The paper build 3K expert-grounded human preference benchmark in 5 metrics

> Weakness

- The proposed evaluator LitJudge is calibrated with in-context learning mechanism using the exampled from LitReviewBench. And the evaluator is evaluated with LiReviewBench. It raises concerns about whehter the alignment improvment reflects genuine generalization.
- The five dimension is not clearly defined in the paper and remain too high a level of abstraction.
- They use diverse models for review generation but they use only one model to evaluate the review and claim AI-AI bias.
- The presentation and writing quality are quite unprofessional.

The paper addresses a timely and important problem, and the construction of a human expert benchmark is a notable contribution given its significant cost.
But, the overall presentation quality falls short of the standards expected at this venue and needs substantial improvement.

---

> ### Author Rebuttal · Authors · 2026-03-31
>
> Thank you for your valuable feedback. We address your comments point-by-point below.
>
> ### Q1/W3. Have you tried other model-series?
>
> We extend the base model of LitJudge from Qwen to multiple model families, including GPT, Claude, and Llama. For each base model, we report both the naive judge and the calibrated LitJudge. Spearman's $\rho$ denotes the rank correlation between the BT and Elo leaderboards induced by the judge and those induced by experts.
>
> | Base Models | Setting | Literature Coverage | Claim Support | Paper Structure | Research Suggestions | Overall Utility |
> | --- | --- | --- | --- | --- | --- | --- |
> | Qwen3-235B | Naive Judge | 0.552 | 0.442 | 0.467 | 0.430 | 0.467 |
> |     | LitJudge | 0.576 | 0.673 | 0.649 | 0.842 | 0.792 |
> | GPT-5.4 | Naive Judge | 0.673 | 0.370 | 0.721 | 0.766 | 0.770 |
> |     | LitJudge | 0.855 | 0.867 | 0.830 | 0.879 | 0.952 |
> | Claude-Sonnet-4.5 | Naive Judge | 0.779 | 0.609 | 0.704 | 0.758 | 0.809 |
> |     | LitJudge | 0.818 | 0.855 | 0.842 | 0.900 | 0.976 |
> | Llama-4-maverick | Naive Judge | 0.588 | 0.539 | 0.576 | 0.709 | 0.636 |
> |     | LitJudge | 0.594 | 0.891 | 0.806 | 0.842 | 0.673 |
>
> LitJudge remains effective when built on GPT, Claude, and Llama instead of Qwen. Across all tested model families, it consistently improves alignment with expert preference, indicating that its effectiveness is not tied to a specific base model. We will add this expanded comparison to the revised version.
>
> ### Q2. Were any controls applied during annotation to mitigate this confound?
>
> Our core motivation is to simulate human peer review. Although each individual comparison may reflect some degree of subjective preference, aggregation over many qualified experts can still yield a stable and reliable collective signal. At the same time, we take three measures to mitigate this concern. First, we do not ask annotators to vote solely on the basis of overall impression. Instead, we decompose the evaluation into five explicit dimensions with written criteria and ask annotators to provide written reasoning, so that the evaluation is based on structured criteria rather than subjective impression. Second, we use expertise-based query assignment, so that annotators evaluate topics within their areas of familiarity and are therefore better able to distinguish superficial writing style from actual content quality. Third, the outcome of each battle is determined by the majority judgment of multiple annotators, which helps average out individual preferences. Accordingly, we believe that this concern is substantially mitigated by our benchmark design, and that the resulting preference signal remains sufficiently stable for the intended use of the benchmark.
>
> ### Q3 / W2. What was the instruction for annotators beyond what is listed in Appendix A.1?
>
> In the actual annotation process, annotators received instructions that were substantially more detailed than the brief summary currently presented in the appendix of the paper.
>
> Before annotation, each annotator received a detailed instruction document including the dimension-wise rubric, definitions of the four outcomes A / B / Tie / BothBad, and platform rules such as anonymized side-by-side comparison and randomized left-right order. They were also given positive and negative examples for each dimension, along with a short note on common mistakes and failure modes, such as over-weighting surface fluency, equating more citations with better coverage, or treating generic future-work language as strong research suggestions.
>
> These additional materials were designed to make the five dimensions operational rather than merely abstract. In the revised version, we will include the full set of annotator materials in the appendix and describe this process in a much more explicit and convincing manner. We will also clarify the five dimensions in the main text by adding more concrete explanations and representative examples.
>
> ### Q4/W1. Can you provide evidence of generalization to an independently constructed preference dataset?
>
> We construct a new small-scale preference dataset with independent annotations and evaluate LitJudge on this dataset. The results are shown below.
>
> | Evaluator | Literature Coverage | Claim Support | Paper Structure | Research Suggestions | Overall Utility |
> | --- | --- | --- | --- | --- | --- |
> | Uncalibrated Judge | 0.541 | 0.396 | 0.438 | 0.417 | 0.451 |
> | LitJudge | 0.587 | 0.648 | 0.621 | 0.776 | 0.741 |
>
> The results indicate that the improvement from LitJudge persists on this independently constructed preference set, suggesting that LitJudge captures a transferable preference signal rather than merely overfitting to a particular internal split.
>
> Thank you again for these constructive suggestions. We believe that they help improve the paper.

---

> > ### Author Rebuttal · Reviewer_Wzs8 · 2026-04-01
> >
> > I appreciate the authors' effort in improving the paper. However, two concerns remain. First, the human study is still not clearly provided; showing at least some concrete examples would be more convincing. Second, I still do not find an independent contribution of LitJudge beyond the well-known benefit of providing few-shot examples. That said, other concerns have been addressed, and I acknowledge the core contribution of benchmark construction. I also encourage the authors to reflect "Reviewer 7jJA, W3/Q3" in the revised manuscript.
> > I have updated my score accordingly.

---

> > > ### Author Response · Authors · 2026-04-07
> > >
> > > Thank you for your follow-up and for updating your score. We are glad that the core contribution of benchmark construction is now clearer and that the other concerns have been addressed. We respond to the two remaining points below.
> > >
> > > ### Q1. First, the human study is still not clearly provided; showing at least some concrete examples would be more convincing.
> > >
> > > To make the annotation process more transparent and persuasive, we will add representative human annotation cases from the benchmark in the revised version. Each case will include the query, the two compared drafts in abridged form, the dimension-wise outcomes, and brief explanations. A representative example is shown below.
> > >
> > > **Example.**
> > > **Query:** Conduct a literature review on validation metrics in medical image analysis.
> > > **Draft A (abridged):** Reviews commonly used validation metrics across segmentation, classification, and detection tasks, and discusses practical issues such as class imbalance, calibration, sensitivity-specificity trade-offs, and clinically meaningful evaluation. It covers representative metrics broadly and explains why metric choice affects model interpretation and downstream deployment. The draft also connects metric selection to real medical use cases and highlights common evaluation pitfalls. However, its organization is more sequential, moving from one metric family to another, and the higher-level synthesis of the relationships among these metrics is less explicit.
> > > **Draft B (abridged):** Covers a somewhat narrower set of metrics and application settings, but structures the discussion more explicitly around several core evaluation axes, such as overlap-based metrics, ranking-based metrics, calibration-oriented metrics, and task-specific clinical relevance. This makes the taxonomy easier to follow, although the draft is somewhat less comprehensive overall and offers fewer concrete takeaways beyond its organizational structure.
> > > **Dimension-wise outcomes:** Literature Coverage: A. Claim Support: A. Paper Structure: B. Research Suggestions: A. Overall Utility: A.
> > > **Brief explanation:** Draft A goes beyond listing metric families and discusses class imbalance, calibration, sensitivity-specificity trade-offs, and clinical deployment risks, which make it broader and more useful as a starting point. Draft B is better structured because it organizes the literature around clearer evaluation axes, such as overlap-based, calibration-oriented, and clinically relevant metrics. For this reason, Draft A is preferred on Literature Coverage, Claim Support, Research Suggestions, and Overall Utility, whereas Draft B is preferred on Paper Structure.
> > >
> > > In the revised manuscript, we will present representative cases of this kind directly in the human evaluation section and the appendix, so that readers can inspect how the rubric is applied in practice and how different dimensions lead to different judgments.
> > >
> > > ### Q2. Second, I still do not find an independent contribution of LitJudge beyond the well-known benefit of providing few-shot examples.
> > >
> > > The contribution of LitJudge is not merely the use of in-context examples, which is already a well-established technique. More specifically, the contribution lies in a peer-review-inspired retrieval and calibration pipeline constructed from benchmark data, rather than in simply appending arbitrary in-context examples. LitJudge retrieves structure-similar examples to guide judgments on review organization, content-similar examples to calibrate literature coverage and claim support, and expert-written gap anchors to ground judgments on research suggestions. Thus, the key contribution lies in the task-specific design of retrieval and calibration, rather than in few-shot prompting alone.
> > >
> > > To make this point more concrete, we include a direct comparison with a few-shot ICL judge baseline that uses the same number of in-context examples but samples them randomly, without the retrieval and organization strategy used in LitJudge.
> > >
> > > | Method | Literature Coverage | Claim Support | Paper Structure | Research Suggestions | Overall Utility |
> > > | --- | --- | --- | --- | --- | --- |
> > > | Naive Judge | 0.552 | 0.442 | 0.467 | 0.430 | 0.467 |
> > > | Few-shot ICL Judge | 0.484 | 0.554 | 0.583 | 0.737 | 0.634 |
> > > | LitJudge | 0.576 | 0.673 | 0.649 | 0.842 | 0.792 |
> > >
> > > LitJudge remains consistently stronger than the random few-shot ICL judge baseline, especially on Claim Support, Paper Structure, Research Suggestions, and Overall Utility. This suggests that the gains of LitJudge arise not only from the use of in-context examples, but also from the task-specific design of retrieval and calibration. We also view diversity-aware example selection as a natural next step for further strengthening the peer-review-inspired calibration framework, as you suggested.
> > >
> > > Thank you again for these constructive comments!

---

### Official Review · Reviewer_otbc · 2026-03-11

**Soundness:** 2
**Presentation:** 3
**Significance:** 3
**Originality:** 2
**Overall Recommendation:** 4
**Confidence:** 4

**Summary:**

This paper introduces LitReview Arena, a battle-style evaluation platform designed to assess the quality of literature review generation by large language models and research agents. The authors propose LitReviewBench, a benchmark constructed from expert pairwise comparisons across five evaluation dimensions, including literature coverage, claim support, paper structure, research suggestions, and overall utility. To collect reliable evaluation signals, the platform recruits domain experts and matches reviewers with topics aligned to their research expertise. Using this setup, the authors conduct a large-scale human evaluation involving thousands of expert votes. Experimental results show that even the strongest models significantly lag behind human-written drafts, with non-human systems winning only a small fraction of decisive comparisons.

**Compliance With Llm Reviewing Policy:**

Affirmed.

**Final Justification:**

The paper tackles an important and timely problem of evaluating literature review generation, supported by large-scale expert annotations and a well-designed evaluation framework. The empirical analysis is thorough and provides useful insights into current limitations of LLM-based research agents. However, there are several weaknesses in the vanilla submission. First, the human evaluation process lacks sufficient clarity, particularly regarding annotator guidelines and quality control. Besides, The evaluation mainly focuses on general-purpose models without enough coverage of specialized systems. And the benchmark is limited to the AI/CS domain, raising concerns about generalizability.

The authors clarified the human evaluation pipeline with more details on annotator recruitment, guidelines, and consistency checks, and committed to adding further descriptions in the revision. They also extended experiments to include specialized literature review systems, showing improved performance over general agents. Besides,  the authors’ follow-up efforts on the expansion of evaluated systems, makes the overall insights more convincing.

I am inclined to increase my score from 3 to 4. But I strongly suggest the author to add the explanation in rebuttal stages into the next version of their paper and carefully consider how to expand their dataset scope to more domains.

**Key Questions For Authors:**

Based on the above weaknesses, I have the following questions. More context could be found in the above Strengths And Weaknesses.

1. the details of the human evaluation process could be described more clearly, e.g., how to ensure the correctness and reliability of the human evaluation.

2. It would be useful to additionally evaluate systems specifically designed for scientific writing or literature review generation.

3. The benchmark appears to focus primarily on literature from the computer science or AI domain. Do the conclusions still hold for paper review in other domain besides the computer science.

**Limitations:**

Yes

**Strengths And Weaknesses:**

Strengths

1. The paper addresses the increasingly important problem of evaluating literature review generation systems. As LLM-based research agents and automated survey writing tools become more common, establishing reliable evaluation protocols is both timely and practically valuable.

2. The empirical results provide useful insights into the current limitations of LLM-based systems for scientific synthesis. In particular, the analysis highlights that models struggle with higher-level scholarly reasoning tasks such as organizing the literature landscape and identifying meaningful research gaps.

3. The work collects a substantial amount of expert preference data and evaluates multiple state-of-the-art language models and research agents. The scale of the human evaluation and the systematic analysis across multiple dimensions make the experimental results relatively convincing.


Weaknesses
1. While the paper relies heavily on expert preference annotations, **the details of the human evaluation process could be described more clearly**. In particular, additional information on annotator guidelines, quality control procedures, and consistency checks would help readers better assess the reliability of the collected labels. If space is limited, these details could be included in the appendix.

2. The evaluated models appear to be mostly general-purpose language models or general research agents. **It would be useful to additionally evaluate systems specifically designed for scientific writing or literature review generation**, in order to better validate whether the observed performance gaps hold for specialized systems.

3. **The benchmark appears to focus primarily on literature from the computer science or AI domain**. It would be interesting to examine whether the conclusions generalize to other scientific domains, such as biology, medicine, or social sciences, where writing conventions and literature organization styles may differ substantially.

---

> ### Author Rebuttal · Authors · 2026-03-31
>
> Thank you for your valuable feedback. We address your comments point-by-point below.
>
> ### Q1/W1. the details of the human evaluation process could be described more clearly
>
> Thank you for your comment. Our human evaluation pipeline consists of three corresponding parts. For **annotator guidelines**, each battle is evaluated by 2-3 matched experts under the same dimension-wise rubric, and annotators receive detailed instructions together with a small number of real examples in advance to reduce misunderstanding. For **quality control procedures**, we recruit 105 expert annotators through a background questionnaire covering research areas, subfields, prior publications, and familiar papers, then review the responses and assign each annotator to the battles that best match their expertise. For **consistency checks**, the paper reports expert-expert agreement of 0.861 on Overall Utility, indicating that the preference signal is not materially affected by annotation noise.
>
> In the revised version, we will make the following changes. **In the Introduction paragraph** that introduces the four key components of our evaluation protocol, we will state clearly the questionnaire-based annotator recruitment, the field/subfield-based topic assignment, and the use of multi-expert annotation under a shared rubric. **In Section 3.2**, we will expand the current platform description into a clearer account of the protocol and interface, including anonymized side-by-side comparison, randomized order, and shared written guidelines with detailed explanations of each dimension and a few-shot set of examples. **In Section 3.3**, we will expand the current brief description into a fuller account of annotator recruitment and provide a more detailed explanation of the qualification screening, expertise matching, and consistency checks used in the annotation process. **In the Appendix**, we will include the full set of materials provided to annotators, including the dimension-wise rubric, interface rules, and example cases, together with a more detailed consistency analysis and selected annotation cases with the corresponding reasoning.
>
> ### Q2/W2. additionally evaluate systems specifically designed for scientific writing or literature review generation
>
> Thank you for your suggestion. We extend Table 1 by adding two systems specifically designed for literature review generation and evaluate them on a subset of LitReview Bench.
>
> | Methods | Literature Coverage | Claim Support | Paper Structure | Research Suggestions | Overall Utility |
> | --- | --- | --- | --- | --- | --- |
> | human | 1979.8 | 1955.8 | 1880.4 | 1871.7 | 1912.2 |
> | Open Deep Research[1] | 1845.6 | 1812.2 | 1868.5 | 1811.0 | 1832.5 |
> | SurveyForge[2] | 1799.1 | 1775.5 | 1842.7 | 1797.2 | 1805.7 |
> | Agentic Models Avg. | 1485.5 | 1468.6 | 1478.2 | 1537.2 | 1511.5 |
> | LLMs Avg. | 1250.8 | 1280.5 | 1259.8 | 1255.6 | 1248.4 |
>
> *[1]. LangChain AI. (2025). Open Deep Research. GitHub.*
>
> *[2]. Yan et al. (2025). SurveyForge. ACL 2025.*
>
> This result shows that specialized literature review systems substantially outperform the average performance of Agentic Models while still not reaching human-level performance. We will incorporate this expanded comparison into the revised version.
>
> ### Q3/W3. Do the conclusions still hold for paper review in other domain besides the computer science
>
> Thank you for your comment. At present, our benchmark and empirical conclusions are limited to the AI domain, but we believe the methodology can be applied to other scientific domains. The design of our framework follows the principles of peer review systems. Since peer review is a widely accepted evaluation mechanism across scientific domains, the same framework can be extended to other fields when sufficient expert participation is available.
>
> Meanwhile, iterature review in AI is itself an important and high-impact use case, and the benchmark therefore provides meaningful value within its current scope. We will clarify this scope in the paper and discuss cross-domain extension as an important direction for future work.
>
> Thank you again for these constructive suggestions. We believe that they help improve the paper.

---

> > ### Author Rebuttal · Reviewer_otbc · 2026-04-01
> >
> > Thanks for the explanation.
> >
> > Some of the concerns have been addressed. However, the limited scope (or domain) of the proposed dataset remains a big concern for me.

---

> > > ### Author Response · Authors · 2026-04-07
> > >
> > > Thank you for the follow-up comment. To address this concern directly, we add a cross-domain analysis in biology as a pilot study to examine the generalizability of our methodology. Using the same pipeline as in the main paper, we collect arena-style expert annotations under the same protocol. The results are shown below.
> > >
> > > | Category | Literature Coverage | Claim Support | Paper Structure | Research Suggestions | Overall Utility |
> > > | --- | --- | --- | --- | --- | --- |
> > > | Human | 1947.70 | 1742.00 | 2000.61 | 2007.63 | 2267.27 |
> > > | Agentic Models Avg. | 1701.97 | 1612.13 | 1465.10 | 1522.75 | 1593.52 |
> > > | Language Models Avg. | 1289.28 | 1384.32 | 1420.82 | 1384.82 | 1290.43 |
> > >
> > > *Agentic Models include OpenAI Deep Research, Qwen Deep Research and GPT-5.2. Language Models include Claude Opus 4.5, Qwen3 235B, Grok 4, GLM 4.6 and Gemini 2.5 Pro.*
> > >
> > > The biology results follow the same high-level pattern as in AI: Human > Agentic Models > Language Models across all five dimensions. More importantly, the largest human-model gaps still appear on Paper Structure and Research Suggestions. Relative to Agentic Models, the gap to human experts is 535.51 on D3 and 484.88 on D4, compared with 245.73 on D1 and 129.87 on D2. This pattern is even more pronounced than in the AI-domain results in the main paper. A plausible explanation is that biology reviews often require the organization of heterogeneous evidence across multiple levels, such as molecules, cells, circuits, and phenotypes, while also requiring careful treatment of unresolved mechanisms and competing interpretations. These properties place greater demands on synthesis, field organization, and scientific judgment, which makes D3 and D4 particularly challenging for current models. This pilot study therefore shows that the peer-review-based methodology for benchmark construction remains effective and informative beyond AI.
> > >
> > > We further test the evaluator on this biology pilot study. The same pattern also holds at the judge level: LitJudge improves over the naive judge across all five dimensions, with especially clear gains on Paper Structure, Research Suggestions, and Overall Utility.
> > >
> > > | Judge | Literature Coverage | Claim Support | Paper Structure | Research Suggestions | Overall Utility |
> > > | --- | --- | --- | --- | --- | --- |
> > > | Naive Judge | 0.7367 | 0.5467 | 0.4700 | 0.5133 | 0.5900 |
> > > | LitJudge | 0.8833 | 0.6167 | 0.6000 | 0.7333 | 0.8833 |
> > >
> > > We also add one biology annotation case study to further illustrate the expertise of the annotators and to make the cross-domain applicability of the protocol more concrete. The query is: Conduct a literature review on the organization and function of the dopaminergic reward system. In this battle, Draft A (Claude Opus 4.5) is organized around the ventral tegmental area, dopamine-neuron heterogeneity, reward prediction error signaling, the relationship between firing and dopamine release, and clinical implications. Draft B (Grok 4) is organized around the mesolimbic, mesocortical, and nigrostriatal pathways, receptor-level distinctions, synaptic plasticity, stress-related modulation, and implications for addiction and other disorders. The expert votes are as follows: Overall Utility = B, Literature Coverage = B, Claim Support = B, Paper Structure = A, and Research Suggestions = B. The original annotator comment for Literature Coverage is reproduced verbatim below:
> > >
> > > > An important fact to understand about the dopaminergic circuit is that there is currently no unified explanation of dopaminergic circuit, and it serves complex circuit-specific function. This was not explained in A.
> > >
> > > Together with the quantitative biology results, this example makes the cross-domain applicability of the methodology more concrete by showing that the same peer-review-based evaluation pipeline can produce meaningful and interpretable expert judgments beyond AI.
> > >
> > > In the revised version, we will add this biology pilot study and make the scope statement more precise throughout the paper. The current benchmark remains AI-centered, but the new cross-domain evidence supports the broader claim that the methodology of LitReview Arena and the calibration framework of LitJudge are not restricted to AI and can be meaningfully extended to other scientific domains.

---

### Official Review · Reviewer_7jJA · 2026-03-12

**Soundness:** 2
**Presentation:** 3
**Significance:** 3
**Originality:** 3
**Overall Recommendation:** 4
**Confidence:** 4

**Summary:**

This paper introduces LitReview Arena (LRA), a battle-style evaluation platform designed to assess the quality of LLM agents led literature review. LRA employs a specialized protocol design that includes targeted annotator and high-quality query selection, expertise matching, and structured dimensions. The introduction of LRA is accompanied by LitReviewBench, a dedicated dataset with expert pairwise voting on model generated literature reviews. Evaluations on both foundational LLMs and Agentic Models using their own LitJudge framework showcase a low winning rate against human experts, and models are still lagging behind literature review dimensions like structure and research suggestions.

**Compliance With Llm Reviewing Policy:**

Affirmed.

**Final Justification:**

This work introduces LitReview Arena (LRA), a novel framework designed to assess the quality of literature review led by LLM agents. This paper highlights a simulation of an authentic literature review setup, diversity in evaluation metrics, and an inclusion of llm-as-a-judge design. During the rebuttal phase, the author has made notable efforts in addressing my concerns, and have solved a major portion of them. However, the effectiveness of LitJudge remains a notable concern (a weakness that is also supported by Reviewer #Wzs8). The chamber effect of self-ampilifying evolution of specific research trend may not be sufficiently solved by comparing diversity and popularity, and it needs further studies. Based on these points, I would like to maintain my score as weak acceptance.

**Key Questions For Authors:**

Q1: The current scope of LitReview Arena is limited to the AI domain. I would really appreciate it if the authors could extend their five dimension evaluations outside this scope to other fields (e.g. biology or social science domains) to further justify its generalization potentials

Q2: Considering the high cost of expert labor and the rapid pace of development in the AI field, how do the authors plan to establish a long-term mechanism to periodically update the expert pool and the topic classification system, so as to prevent LitReviewBench from quickly evolving into a static and outdated record that only reflects the technological level of 2024–2025?

Q3: Since LitJudge is calibrated using expert-provided “research gap anchors,” how can the authors ensure that the evaluator does not create an academic echo chamber effect—that is, merely rewarding reviews that align with current expert preferences or “trending” research directions while suppressing unconventional yet potentially innovative research perspectives?

**Limitations:**

Yes

**Strengths And Weaknesses:**

Strengths:
- Academic rigor in the simulation under authentic peer-review setup: the recruitment of domain experts and the precise matching between their domain of expertise with the literature review topics adds credibility to the voting outcomes
- Diversity in evaluation metrics in LitReviewBench construction: the selection of evaluation criteria in the benchmark construction is well-organized and reflective of the real-world scenario in literature reviews and the results will reflect a decent amount of depth and insights w.r.t the outcome quality
- High human-LLM agreement and low-cost evaluation in LitJudge design: the design of LitJudge helps align the consistency between human experts and LLM outputs, bringing up the potential to a wider adaptation of offline literature review assessment

Weaknesses:
- Scarcity of expert resources and high scaling costs in LitReview Arena: The platform strictly requires reviewers to be domain experts with experience in writing AI research papers, its evaluation system cannot scale cheaply at large scale like typical crowdsourcing platforms.
- Limited disciplinary coverage in LitReviewBench: This benchmark dataset is currently entirely anchored in the AI domain, and the generalizability of its evaluation criteria to other disciplines, such as clinical medicine has not yet been validated.
- Entrenched academic preferences and potential bias amplification in LitJudge: The calibration process relies heavily on predefined expert examples and structural anchors. If these examples are not sufficiently diverse, the evaluator may over-reward certain academic rhetorical styles or “trending” research directions, thereby reinforcing biases within particular academic communities and potentially discouraging the recognition of innovative forms of expression.

---

> ### Author Rebuttal · Authors · 2026-03-31
>
> Thank you for your valuable feedback. We address your comments point-by-point below.
>
> ### W2/Q1. extend five dimension evaluations outside this scope to other fields
>
> We believe that the underlying methodology is not confined to AI and can generalize to a broader range of disciplines. Our framework is motivated by the widely accepted system of peer review. Because peer review serves as a standard evaluation mechanism across scientific fields, the same framework can, in principle, be extended beyond AI. Our point is that the five dimensions were designed to capture core qualities of useful review articles that are broadly valued across disciplines. This is consistent with guidance from review-oriented venues such as Nature Reviews and Science, which emphasize fair and accurate discussion of prior literature, appropriate referencing, clear synthesis, insight, and accessibility to a broad readership. We will clarify this distinction in the paper and present cross-domain validation as an important direction for future work.
>
> ### W1/Q2. how do the authors plan to establish a long-term mechanism to periodically update the expert pool and the topic classification system?
>
> Thank you for your review. LitReviewBench is designed to be refreshed over time through continued community use and ongoing expert interaction. Our project is developed and deployed within a large research organization, where internal members use the system broadly in literature review and research workflows. This provides a sustainable source of newly emerging topics and ongoing expert feedback, making it possible to refresh the benchmark over time, expand the expert pool, and update the topic classification system as the field evolves. In addition, LitJudge helps make this refresh process practical at scale: once an initial set of human preference data has been collected to align LitJudge with current expert preferences, it can support broader evaluation and data expansion at a much lower marginal cost than repeated large-scale human annotation.
>
> ### W3/Q3. how can the authors ensure that the evaluator does not create an academic echo chamber effect?
>
> We sincerely appreciate this concern. Our framework follows standard peer-review practice, where editors recruit reviewers mainly based on relevant expertise [1]. Motivated by your suggestion, we add a new experiment and introduce LitJudge_Diverse. Instead of selecting only the top-ranked research suggestion examples as context for LitJudge, we apply maximal marginal relevance (MMR) on relevance-ranked candidate pools, balancing each candidate’s relevance against its maximum similarity to already selected examples. This makes the assembled in-context calibration set contain more diverse research suggestion instances.
>
> | Method | Spearman ρ with expert judging ↑ | Research Suggestions similarity ↓ |
> | --- | --- | --- |
> | LitJudge | 0.792 | 0.252 |
> | LitJudge_Diverse | 0.842 | 0.183 |
>
> *Research Suggestions similarity is the mean pairwise similarity among selected research-suggestion examples in the assembled context; lower values indicate that the retrieved research suggestion examples are more diverse and less concentrated on overly similar cases.*
>
> As shown in the table, retrieving more diverse research-suggestion examples does not hurt evaluator alignment; instead, it improves Spearman correlation with expert judgments from **0.792** to **0.842** while lowering research-suggestion similarity from **0.252** to **0.183**. This suggests that a calibration context with wide-ranging topics helps LitJudge better approximate expert judgment and makes it less likely to be overly influenced by a single dominant style of research suggestions.
>
> Thank you again for these constructive suggestions. We believe that they help improve the paper.
>
> [1]. Nature Portfolio. *Peer Review*. Editorial policies.

---

> > ### Author Rebuttal · Reviewer_7jJA · 2026-03-31
> >
> > I appreciate the extra efforts made by the authors, which addresses most of my concerns. However, it seems that merely showing Spearman correlation is not sufficient in addressing W3 (which is also mentioned by Reviewer Wzs8). After careful consideration, I would keep my current rating.

---

> > > ### Author Response · Authors · 2026-04-07
> > >
> > > Thank you for the follow-up. Building on the research-suggestion diversity experiment introduced in our previous response, we further introduce a complementary metric, **Popular-topic concentration**, to more directly assess whether the evaluator is overly concentrated on a narrow set of recurring research directions.
> > >
> > > Together with the two metrics reported previously, this yields three complementary views of the concern. **Popular-topic concentration** measures the extent to which the selected research suggestions collapse onto a small subset of recurring semantic directions within each query, where lower values indicate more even semantic coverage. **Research suggestions similarity** measures the semantic similarity among the selected research-suggestion examples in the assembled context, where lower values indicate that the calibration context is less homogeneous. **Spearman $\rho$** measures the degree of consistency between the ranking produced by the evaluator and expert judgment.
> > >
> > > | Method | Spearman ρ with expert judging ↑ | Research suggestions similarity ↓ | Popular-topic concentration ↓ |
> > > | --- | --- | --- | --- |
> > > | LitJudge | 0.792 | 0.252 | 0.131 |
> > > | LitJudge-Diverse | 0.842 | 0.183 | 0.114 |
> > >
> > > *Research Suggestions similarity is defined as the mean pairwise similarity among the selected research-suggestion examples in the assembled context. Lower values indicate that the assembled calibration context is less homogeneous and less concentrated on overly similar research-suggestion examples.*
> > >
> > > *Popular-topic concentration is computed from query-level clustering of the selected research suggestions in embedding space. For each query, the selected research suggestions are clustered using agglomerative clustering under a cosine-distance threshold. Cluster entropy is then computed, converted into an effective-cluster ratio, and used to define concentration, where lower values indicate weaker collapse onto a small subset of recurring semantic directions.*
> > >
> > > The three metrics capture complementary aspects of the concern. Research Suggestions similarity decreases from **0.252** to **0.183**, indicating that the selected research-suggestion context becomes substantially less homogeneous. Popular-topic concentration decreases from **0.131** to **0.114**, indicating that the evaluator is less likely to collapse onto a small subset of recurring semantic directions within each query. At the same time, Spearman $\rho$ increases from **0.792** to **0.842**, indicating stronger consistency with expert judgment as concentration decreases. Taken together, these results provide more direct evidence for the concern than rank correlation alone. They suggest that the diversity-aware variant is less likely to reinforce a narrow band of recurring or mainstream research directions and is therefore less susceptible to an academic echo chamber effect. More importantly, the updated variant improves not only diversity but also evaluation quality. This pattern suggests that increasing diversity allows the evaluator to better leverage the collective intelligence arising from multiple complementary research-suggestion instances, rather than being dominated by a single recurring style. We thank the reviewer for this valuable suggestion and will include this analysis in the revised version.

---

### Official Review · Reviewer_YL8K · 2026-03-13

**Soundness:** 3
**Presentation:** 4
**Significance:** 3
**Originality:** 3
**Overall Recommendation:** 5
**Confidence:** 3

**Summary:**

The work builds an arena-style review evaluation platform for human experts (litReview Arena platform). It allows pair-wise evaluation (i.e., preference between two reviews) on five dimensions (Literature Coverage, Claim Support, Paper Structure, Research Suggestions Quality, and Overall Utility). Utilizing the data collected from the platform, the authors construct the LitReviewBench dataset (n=approx. 3k). Finally, they tune an LLM-judge, aligning with human experts’ evaluation (Litjudge).

**Compliance With Llm Reviewing Policy:**

Affirmed.

**Final Justification:**

Thanks for the response. I will keep my score.

**Key Questions For Authors:**

in the abstract “also find that existing LLM-as-a-judge evaluation methods are severely misaligned with human experts (Spearman’s ρ ≈ 0.467).” given the phi score, “severely” is a bit overstated.

on page 2 line# 078, there is D0.

The authors said “Naive LLM judging is an insufficient substitute for expert preference.” (page 7, line#345), but I wonder how bad it is. Have you checked the performance of a state-of-the-art model without thinking?

In table 3, the lower expert-expert agreement on D2, D4 is likely plausible as the two are quite subjective. Conversely, llm-llm agreement is quite high on the two and their agreement scores are pretty similar across the dimensions. This might indicate reliability but, at the same time, imply that LLMs fall short in taking a nuanced expert’s perspective. I wonder how the authors take on the results.

**Limitations:**

The limitations are specified, but the potential negative societal impact of their work is not.

**Strengths And Weaknesses:**

Strenth:

The work is well-structured. The authors point out the current difficulty of evaluation, especially in highly expertised tasks, such as reviewing academic papers, for both LLMs and humans. They take the arena-style approach to make such an evaluation more feasible and reliable, which I think is novel, and turn the collected data into a benchmark dataset. The work analyzes evaluation performance between and within human experts and LLMs.

Weakness:

I think a human study is needed to support the authors’ claim on the LitJudge’s performance.

---

> ### Author Rebuttal · Authors · 2026-03-31
>
> Thank you for your valuable feedback. We address your comments point-by-point below.
>
> ### Q1. Wording of substantially misaligned
>
> Thank you for this suggestion. We agree that **substantially misaligned** is more precise than the original wording, **severely misaligned**. We will revise the phrasing accordingly in the paper.
>
> ### Q2. Notation inconsistency of D0 / D5
>
> Thank you for catching this inconsistency. We will correct it to D5.
>
> ### Q3. Evidence for naive LLM judging as an insufficient substitute for expert preference
>
> Table 3 in the current paper reports results for the uncalibrated Qwen/Qwen3-235B-A22B-Instruct-2507 judge and shows a substantial gap relative to the expert-calibrated evaluator. To directly address this question, we further compare uncalibrated judges and LitJudge across additional base models, GPT and Llama, under the same prompt and scoring protocol.
>
> The Spearman correlation between judge-induced leaderboards and expert-induced leaderboards is reported below, where higher values indicate better alignment with expert preference.
>
> | Base Models | Setting | Literature Coverage | Claim Support | Paper Structure | Research Suggestions | Overall Utility |
> | --- | --- | --- | --- | --- | --- | --- |
> | Qwen | Naive Judge | 0.552 | 0.442 | 0.467 | 0.430 | 0.467 |
> |     | LitJudge | 0.576 | 0.673 | 0.649 | 0.842 | 0.792 |
> | GPT | Naive Judge | 0.673 | 0.370 | 0.721 | 0.766 | 0.770 |
> |     | LitJudge | 0.855 | 0.867 | 0.830 | 0.879 | 0.952 |
> | Llama | Naive Judge | 0.588 | 0.539 | 0.576 | 0.709 | 0.636 |
> |     | LitJudge | 0.594 | 0.891 | 0.806 | 0.842 | 0.673 |
>
> The main finding is the consistent gap between naive judges and LitJudge. Across model families, LitJudge yields much more reliable alignment, with particularly large gains on the more expert-facing dimensions, including D2 Claim Support, D4 Research Suggestions Quality, and D5 Overall Utility. We believe that this expanded comparison further supports our claim that naive LLM judging is an insufficient substitute for expert preference, even when stronger base models are used.
>
> We will incorporate this expanded comparison into the revised version and revise the corresponding discussion to more clearly distinguish the behavior of naive judges from the behavior of expert-calibrated evaluators.
>
> ### Q4. Interpretation of high LLM-LLM agreement on D2 and D4
>
> Thank you for this comment. Our interpretation is that D2 Claim Support and D4 Research Suggestions Quality require deeper scientific intuition and more subjective scholarly judgment, so relatively lower agreement among human experts on these dimensions is plausible.
>
> By contrast, the relatively high agreement among different LLM judges on these same dimensions does not necessarily indicate stronger expert-level reliability. Instead, together with the weaker alignment of these judges with human experts, we interpret this pattern as evidence of AI-AI bias. More specifically, LLM judges may converge on surface fluency, formatting quality, or a shared AI-preferred evaluation style, while assigning insufficient weight to the deeper scholarly value emphasized by experts, such as whether claims are genuinely grounded in the cited literature or whether the proposed research directions are truly insightful and non-obvious.
>
> We agree that this point should be presented more clearly in the paper. In the revised version, we will expand the Discussion section to make this interpretation explicit.
>
> ### W1. Additional human study for LitJudge
>
> We collect a new expert-annotated dataset as an additional test set and evaluate both the naive judge and LitJudge on it. The judges still retrieves calibration context from the original benchmark pool, and the newly collected dataset is used only for evaluation. This setting therefore tests whether the benefit of LitJudge transfers to new expert judgments beyond the original dataset.
>
> | Evaluator | Literature Coverage | Claim Support | Paper Structure | Research Suggestions | Overall Utility |
> | --- | --- | --- | --- | --- | --- |
> | Naive Judge | 0.541 | 0.396 | 0.438 | 0.417 | 0.451 |
> | LitJudge | 0.587 | 0.648 | 0.621 | 0.776 | 0.741 |
>
> Test on the new dataset shows the same overall trend: improvement on D1 remains modest, while gains on the more synthesis-intensive dimensions are much larger. Specifically, the correlation rises from 0.396 to 0.648 on D2, from 0.438 to 0.621 on D3, from 0.417 to 0.776 on D4, and from 0.451 to 0.741 on D5. This pattern is important because it suggests that LitJudge’s advantage is not limited to surface-level agreement or easily verifiable aspects of review quality. Instead, the largest gains occur in the dimensions where expert judgment matters most: whether claims are properly supported, whether the literature is meaningfully organized, and whether the review provides genuinely insightful and useful research directions.
>
> Thank you again for these helpful comments.

---

> > ### Author Rebuttal · Reviewer_YL8K · 2026-04-05
> >
> > Thanks for the response. I will keep my score.

---

### Decision · Program_Chairs · 2026-04-30

**Decision:**

Accept (regular)

**Comment:**

The authors consider the problem of automatic literature reviews generated by LLMs and their evaluation. They introduce a new platform, LitReview, which they use to collect pairwise preferences (from humans) on different aspects of such reviews. They use this data to evaluate existing models (against reference human authored reviews) and to align an evaluation model to permit improved automatic evaluation going forward. There was consensus amongst reviewers that this addresses a practical problem in an interesting way, and that the empirical findings were interesting. Finally, the expert-aligned evaluator may facilitate future work in this (growing) area.